# OpenReview forum: "Monitoring Monitorability"
_ICML.cc/2026/Conference — ICML 2026 spotlight_

### Official Review · Reviewer_Hxj1 · 2026-03-11

**Soundness:** 4
**Presentation:** 4
**Significance:** 4
**Originality:** 4
**Overall Recommendation:** 6
**Confidence:** 4

**Summary:**

The paper addresses the critical challenge of supervising autonomous AI agents by monitoring their internal reasoning processes, specifically through chain-of-thought (CoT) monitoring. The authors argue that relying solely on external actions or final outputs is insufficient for true interpretability and transparency. To address the fragility of CoT monitoring under varying training regimes, the paper introduced three distinct monitorability evaluation archetypes. The experimental results demonstrate that the proposed approach significantly outperforms traditional action-only monitoring across several benchmarks.

**Compliance With Llm Reviewing Policy:**

Affirmed.

**Final Justification:**

I maintain my Strong Accept recommendation. This paper addresses an important and timely problem: how to supervise increasingly autonomous AI agents by monitoring their internal reasoning processes rather than relying solely on final outputs. The proposed framework for evaluating chain-of-thought monitorability, along with the three evaluation archetypes and associated metrics, provides a structured and measurable approach to a problem that has largely been handled in an ad-hoc manner. The work is technically strong, well-motivated, and likely to have significant impact on future research in agent oversight and interpretability.

In terms of soundness, the methodology is well-defined and supported by extensive experiments across multiple benchmarks. The introduction of quantifiable monitorability metrics and the systematic evaluation framework strengthens the technical rigor. The evaluation appears thorough, and the analysis of different monitoring regimes is convincing.

In terms of originality, the paper makes a novel contribution by formalizing monitorability evaluation into three archetypes and providing a unified measurement framework. This moves the field from qualitative discussions of reasoning transparency to actionable, measurable evaluation protocols.

In terms of significance, the contribution is substantial. As AI systems become more agentic, reliable oversight mechanisms are critical. The proposed framework provides both conceptual clarity and practical tools that other researchers can adopt. The planned release of evaluation code and datasets further increases the paper’s impact and reproducibility.

In terms of clarity, the paper is well organized and easy to follow. The figures were somewhat difficult to read, but the authors committed to improving readability in the final version.

The authors’ rebuttal fully addressed my concerns. They clarified that the three archetypes are intended as a starting taxonomy rather than an exhaustive list, and provided a thoughtful discussion of their limitations and failure modes. My concerns about shared training idiosyncrasies were addressed through justification of using high-capability monitors to estimate an upper bound on monitorability, along with cross-model-family evaluations. The authors also pointed to experiments involving obfuscated chain-of-thought reasoning, demonstrating that the proposed metrics can detect hidden or misleading reasoning. These responses strengthened my confidence in the work.

Overall, this is a technically strong, original, and impactful contribution with well-supported claims and a clear experimental framework. The rebuttal reinforced my positive assessment, and I therefore maintain my Strong Accept recommendation.

**Key Questions For Authors:**

1. You mentioned that OpenAI models were used as the monitor. Given that shared training data or architectural "idiosyncrasies" could artificially inflate monitorability scores, what alternative approaches could you have explored? For instance, would a smaller or  'dumbed-down" monitor or a totally different architecture provide a more "honest" evaluation of the CoT?
2. How do these monitorability metrics hold up if an agent is fine-tuned to "hide" its true reasoning while maintaining a plausible-looking chain of thought?
3. Could you provide a brief analysis of the computational cost of implementing these monitoring archetypes in real-time? Is there a significant trade-off between the depth of monitoring and the latency of the agent?

**Limitations:**

The limitations of this study were appropriately discussed.

**Strengths And Weaknesses:**

Strengths:
- As AI systems migrate from passive models to agentic entities, the need for robust oversight mechanisms is crucial. This paper provides a necessary framework for "white-box" monitoring.
- The introduction of three evaluation archetypes gives readers a structured methodology for a field that has largely been ad-hoc.
- The paper successfully established a suite of quantifiable metrics. This is a significant contribution, because monitorability must be measurable to be seen as actionable in a production or even research environment.


Areas for Improvement
This paper is strong, but there are some areas for improvement:
- Although the three archetypes are well-defined, the paper would benefit from a more exhaustive discussion on why these specific three were chosen and if they cover the full spectrum of agentic behavior. Are there cases where these archetypes might fail?
- The figures and plots contain vital data but are currently difficult to read. Specifically, the font sizes in the figures are too smallt. To ensure the paper is camera-ready  the authors should use bolder fonts and larger labels for all axes, legends, and annotations.

---

> ### Author Rebuttal · Authors · 2026-03-31
>
> Thank you so much for your kind words about our paper!
>
> ___
> Responses to “Areas for Improvement”:
>
> The three archetypes we identified are not meant to be exhaustive. Our goal was to provide an initial eval taxonomy that would serve as a starting point for structuring monitorability evaluations. We imagine there are more archetypes yet to be discovered which will lead to more kinds of monitorability evals!
>
> With regards to archetypal limitations: Process evaluations can fail when the evaluation design assumes the wrong ontology of reasoning steps; for example, if an agent begins using a novel symbolic strategy or effectively “invents new math,” the reasoning may still be monitorable in principle, but not by a process evaluation whose labels were designed around currently expected reasoning patterns. Intervention evaluations require tasks that can naturally be cast into a controlled perturbation format, which may be difficult or unnatural for some realistic settings. Outcome-property evaluations are broader, but can be less diagnostic, since failures may reflect either weak monitorability or weak or noisy outcome label proxies.
>
> We will clarify these points in the paper. Also, thank you for the feedback on figure readability. We will update the figures to be more legible.
> ___
> Responses to your “Key Questions”:
>
> Q1: We agree that alternative approaches would be valuable to pursue. As you point out, one option would be to use smaller or otherwise deliberately weakened monitors, which might reduce shared-idiosyncrasy concerns. However, this comes at the cost of confounding monitorability with monitor capability: there is a tradeoff between using a weaker monitor that may be less exposed to shared training data or architectural idiosyncrasies, and using a stronger monitor that is better at extracting whatever signal is present in the CoT. In this paper, we default to using one of the strongest monitors available (GPT-5) because our goal was to measure the upper envelope of monitorability: if even a strong monitor cannot recover the relevant information from CoT, that is strong evidence that the signal is not readily accessible. We do also include results where we use the OpenAI family of models to monitor non-OpenAI models.
>
> Another option would be to evaluate each agent model using monitors from multiple model families or architectures and report the aggregate results. We will mention these directions in the paper.
>
> Q2. We have already done these experiments (see Appendix H)! We studied model organisms explicitly trained to generate obfuscated CoTs. Those experiments were intended as a direct stress test of this failure mode, and we show that the evaluation suite can detect simple cases in which the model’s visible CoT becomes less faithful because it has been trained to obfuscate.
>
> Note that the gmean-squared metric is also specifically designed to register degradation when the visible CoT becomes less monitorable to the underlying decision process. Thus, if an agent is fine-tuned to hide its true reasoning while producing a plausible-looking CoT, we would generally expect monitorability to fall. For example, for an intervention eval, if the CoT masks evidence of the intervention that is actually driving the behavioral change, TPR should decrease; if it instead introduces plausible but misleading cues suggesting an effect where none exists, TNR should decrease. Either way, the gmean-squared metric would be lower.
>
> Q3. Unfortunately, our paper does not benchmark wall-clock latency or end-to-end real-time overhead for each monitoring archetype. However, it does analyze several closely related compute dimensions: agent inference-time compute, monitor test-time compute, and follow-up questioning, all of which can increase monitoring depth and would likely increase deployment latency in practice.
>
> Figure 1 (left) shows that monitorability tends to increase with reasoning effort / CoT length. Figure 5 shows that increasing monitor-side test-time compute can also improve monitorability. Figure 6 shows that follow-up questioning can further improve monitorability. Finally, Figure 4 makes the compute–monitorability trade-off at matched capability especially explicit: a smaller model run at higher reasoning effort can often be more monitorable, but at higher overall inference compute cost.
> ___
> We would also like to share that will have published an open source version of our metrics code and the following datasets prior to the camera-ready paper deadline:
> - Intervention evals: GPQA, GPQA encoded, AIME, DAFT, Memory,  Scruples, and Sandbagging
> - Process evals: AIME-Math and GPQA-Physics
> - Outcome-property evals: Flaky-Tools (CharXiv subset), ShadeArena, and Impossible Coding Tasks (LiveBench subset)
>
> The other datasets rely on private or restricted data and will be omitted from the release.

---

> > ### Author Rebuttal · Reviewer_Hxj1 · 2026-03-31
> >
> > 1. The authors have provided a clear and sophisticated explanation of the failure modes for their proposed archetypes. This adds necessary nuance to the framework.
> > 2. My concerns regarding agents "hiding" their reasoning were effectively addressed by the pointer to the experiments in Appendix H and the explanation of how the $g$-mean squared metric registers signal degradation in such cases.
> > 3. The authors' justification for using high-capability models (like GPT-5) to establish a "monitorability upper envelope" is sound,
> > 4. and the inclusion of cross-model family evaluations mitigates concerns regarding architectural idiosyncrasies.

---

> > > ### Author Response · Authors · 2026-04-01
> > >
> > > Thanks very much for the prompt acknowledgement!

---

### Official Review · Reviewer_wNyE · 2026-03-13

**Soundness:** 3
**Presentation:** 3
**Significance:** 3
**Originality:** 3
**Overall Recommendation:** 5
**Confidence:** 4

**Summary:**

This paper systematically studies the monitorability of CoT monitoring. Specifically, it proposes an evaluation framework spanning three task categories: Intervention, Process, and Outcome-property, with a new metric, g-mean², for quantifying monitorability. Using this benchmark, the authors evaluate several frontier reasoning models and present a set of insightful findings, such as the advantages of full-trajectory monitoring over action-only or CoT-only monitoring, the tendency for monitorability to improve with longer CoT under increased test-time compute, the improved monitorability of reasoning traces produced by smaller models with higher reasoning effort under comparable capability levels, and the benefits of follow-up questions in eliciting additional reasoning information, *etc*.

**Compliance With Llm Reviewing Policy:**

Affirmed.

**Final Justification:**

The authors' commitment to open-sourcing largely addresses my concerns regarding reproducibility and broader model coverage. As the remaining concerns have also been reasonably responded to, I decide to raise my score to Accept.

**Key Questions For Authors:**

The main questions are already listed in the weaknesses above with clear numbers like ***Q.X***.

**Limitations:**

yes

**Strengths And Weaknesses:**

Overall, I am very positive about this work, and my main suggestions primarily aim to strengthen the robustness and generalizability of the conclusions.

***Strengths:***

- The paper presents a systematic study of CoT monitorability, including benchmark construction, the introduction of a dedicated metric, and insightful analysis.

- The evaluation suite spans diverse task settings and model configurations, leading to several interesting observations about the relationship between reasoning effort, CoT length, and monitorability.

- The paper is clearly written and generally easy to follow.

***Weaknesses:***


Although the paper already includes some results on models beyond the main OpenAI-based setup, I still have the following concerns:

- The experiments rely primarily on closed-source OpenAI models for both agents and monitors. Since the reasoning traces of these models are not fully accessible to other researchers, this creates clear reproducibility challenges and makes it difficult to assess how well the findings generalize across model families. ***Q.1. Could the authors include one more group of results on open-source models to strengthen the empirical coverage and improve reproducibility?***

- When both the agent and the monitor come from the same model family, the results may benefit from a same-family advantage, where the monitor is better able to interpret reasoning patterns that are closer to its own training distribution. ***Q.2. Could the authors provide additional analysis of monitorability in cross-family settings?***

Additionally, I have two concerns regarding the practicality of real-world deployment:

- The monitors used in the experiments, including GPT-5 Thinking, o3, o3-mini, and o4-mini, are themselves large reasoning models, yet the paper provides limited discussion of monitoring cost and deployment overhead. ***Q.3. If the monitor is substantially more expensive than the agent, how practical is this approach in real-world systems?*** It would be helpful to include an analysis of the “weak-to-strong” monitoring setting studied in [1].

- The authors use different prompts for the monitor under different task settings. This raises the question of ***Q.4. whether the results are sensitive to prompt design, and more importantly, whether a unified monitor prompt could be adopted across settings for more realistic deployment.***

Finally, there are a few minor formatting issues.

- The caption of Figure 30 on page 57 contains errors like *Figure ??*, and two subfigures in Figure 32 on page 60 do not display correctly.

[1] Neil Kale et al. Reliable Weak-to-Strong Monitoring of LLM Agents

---

> ### Author Rebuttal · Authors · 2026-03-31
>
> Thanks so much for your review, and for catching those formatting issues!
>
> Responses to your Key Questions:
> ___
>
> Q1+Q2: We agree that broader open-source and cross-family coverage would strengthen both reproducibility and generalization claims. That said, the paper already addresses this to a meaningful degree: beyond OpenAI models, we evaluate other frontier models, with results shown in Figure 1 (right), Figure 3, and Figures 20 and 21 in Appendix G. These experiments were included precisely to broaden empirical coverage beyond OpenAI models and to provide a partial test of cross-family monitorability.
>
> At the same time, we agree that this could be extended substantially further. Our current experiments do not provide a complete open-source or cross-family matrix, and are limited by which models expose sufficiently visible reasoning traces through their APIs. We therefore see broader open-source and cross-family evaluation as an important direction for future work.
>
> We would also like to note that, prior to the camera-ready deadline, we plan to release an open-source version of our metrics code together with a substantial subset of the evaluation suite. Specifically, we expect to release the following datasets:
> - Intervention evals: GPQA, GPQA encoded, AIME, DAFT, Memory,  Scruples, and Sandbagging
> - Process evals: AIME-Math and GPQA-Physics
> - Outcome-property evals: Flaky-Tools (CharXiv subset), ShadeArena, and Impossible Coding Tasks (LiveBench subset)
>
> The remaining datasets rely on private or restricted data and will therefore be omitted from the release. We hope this will make it substantially easier for other researchers to reproduce and extend our analyses on additional model families.
> ___
> Q3: We agree that monitoring cost is important, and that a fuller weak-to-strong analysis would be valuable. While we do not present a full deployment-cost study, the paper already includes a partial analysis of this setting: we evaluate monitors ranging from GPT-5 Thinking down to o4-mini, and study how monitorability varies with monitor strength and monitor test-time compute relative to the agent. In particular, using the mini series to monitor the larger series already gives a meaningful weak-to-strong datapoint, since there is a very sizable difference in both model capability and cost. Our results suggest that even substantially weaker and cheaper monitors can still recover useful signal, and that additional monitor test-time compute can further improve their performance.
>
> More broadly, our paper focuses on measuring and characterizing monitorability rather than deriving the cost-optimal deployment policy. We agree that practical deployment requires an additional analysis of latency, compute, and risk tolerance, which is likely application-dependent. In higher-stakes settings, we expect users to be willing to spend substantially more compute on oversight than in lower-stakes deployments. We view a more explicit cost-effectiveness analysis of weak-to-strong monitoring as an important direction for future work.
> ___
> Q4:   We agree that prompt sensitivity is important. We use different monitor prompts across settings because the evaluations target different properties, and a single prompt would often be misaligned with the behavior being measured. For example, a prompt designed to detect harmful or subversive behavior would not be appropriate for neutral process evaluations that ask whether a reasoning trace contains a particular intermediate property. We therefore chose prompts to match the target property in each evaluation rather than forcing a single formulation across all settings.
>
> That said, our results are not entirely driven by narrowly specialized prompts. For example, the Anti-Scheming monitor prompt is already fairly broad: it asks whether the model is engaging in deceptive, scheming, reward-hacking, or other subversive behavior in its upcoming actions, rather than targeting one specific exploit or task. This suggests that reasonably general prompts can already provide useful signal in some practically relevant settings.
>
> More broadly, we do not think that a single identical prompt across all evaluations is always the most realistic deployment assumption, since monitors in practice are often conditioned on the property of interest. However, we agree that studying prompt sensitivity and more unified prompting schemes is an important direction for future work.

---

> > ### Author Rebuttal · Reviewer_wNyE · 2026-04-03
> >
> > Thanks for the authors' response! I look forward to the open-source release and believe it will be a valuable contribution to the safety community. Based on this, I raise my score accordingly.

---

> > > ### Author Response · Authors · 2026-04-03
> > >
> > > Thank you so much!

---

### Official Review · Reviewer_Wski · 2026-03-13

**Soundness:** 3
**Presentation:** 4
**Significance:** 4
**Originality:** 3
**Overall Recommendation:** 5
**Confidence:** 4

**Summary:**

The paper proposes evaluations of monitorability for LLMs. Specifically, the authors propose a framework to evaluate how well a monitoring system can infer safety-relevant properties from an LLM agent's CoT.. To do this, they introduce a broad evaluation suite consisting of three subsets (intervention, process, and outcome-property evaluations). The authors apply this suite to evaluate several state-of-the-art frontier models (e.g., GPT-5, OpenAI o3, Claude 3.7 Sonnet, DeepSeek R1), analyze scaling trends with respect to pre-training and test-time compute, and demonstrate practical interventions to improve monitorability, such as asking follow-up questions to elicit latent reasoning.

**Compliance With Llm Reviewing Policy:**

Affirmed.

**Key Questions For Authors:**

* Did you do any analysis of whether the proposed evaluations give useful signal about how monitorable a model is in practice?

**Limitations:**

Yes important limitations are discussed in the conclusion section and throughout the paper

**Strengths And Weaknesses:**

**Strengths**
* **Soundness & Methodology**: This is a nice, rigorous approach to measuring monitorability. The evaluation suite is well thought out, covering a diverse set of tasks and behavioral regimes (including sycophancy, bias, misalignment). The choice of the g-mean metric is carefully justified, making the quantitative analysis reliable.
* **Significance**: The problem is clear and well-motivated. The proposed evaluation suite is practically useful and easy for other researchers to build on. The scaling trend analysis is particularly important and well executed, revealing a practical "monitorability tax" (a trade-off between capability and monitorability at fixed compute).
* **Originality**: The formalization of monitorability across three distinct evaluation archetypes provides a novel and comprehensive lens for the community to assess CoT faithfulness and legibility. The paper proposes a novel perspective on the problem of monitorability that goes beyond the common notions of faithfulness and readability.
* **Presentation**: The paper is well written, clearly structured, and easy to follow. The problem is well-motivated, and the figures effectively communicate the main results.

**Weaknesses**
* **Connection to Practical Monitoring**: The biggest limitation is that the paper does not study whether low/high monitorability metrics on these specific evaluations has direct implications for practical monitoring. While the evaluations make sense conceptually, there are many ways they could fail to be predictive. For example, there might be a mismatch between the distribution of tasks in this suite and what is monitored in practice, or the monitors used in the evaluations might not be strong enough to reflect the situation in practical systems.
* **Model-Specific Eval Design**: The evaluations are somewhat specific to the class of models being evaluated. For instance, the intervention datasets can only produce meaningful results if the base models actually respond to the interventions (e.g., hints). This introduces confounders that make it difficult to fairly compare monitorability across different model families and from weaker to stronger, as a model's intrinsic sensitivity to the prompt influences the evaluation.
* **Reproducibility**: There are notable reproducibility issues, in particular for the scaling experiments in Section 5.1. Very few details are provided about the series of pre-trained models used, their architectures, size ranges, or training data. This lack of detail makes it difficult to fully interpret these specific results or compare them directly to public frontier model numbers.

Overall I think this is a strong paper, and think it should be accepted.

---

> ### Author Rebuttal · Authors · 2026-03-31
>
> Thank you so much for the positive words and constructive feedback!
>
> Firstly we would to share that will have published an open source version of our metrics code and the following datasets prior to the camera-ready paper deadline:
> - Intervention evals: GPQA, GPQA encoded, AIME, DAFT, Memory,  Scruples, and Sandbagging
> - Process evals: AIME-Math and GPQA-Physics
> - Outcome-property evals: Flaky-Tools (CharXiv subset), ShadeArena, and Impossible Coding Tasks (LiveBench subset)
>
> The other datasets rely on private or restricted data and will be omitted from the release.
>
> ___
>
> To address your key question, we do provide some evidence that the evaluations capture useful signal relevant to practice. In particular, our obfuscation experiments (Appendix H) show that the suite can detect deliberate degradations of CoT monitorability, which is one concrete failure mode of practical concern. In addition, several evaluations are designed to reflect monitoring-relevant settings where CoT access helps detect problematic behavior better than action-only monitoring (e.g. sandbagging, misalignment (ShadeArena, Repo-Sabotage, Anti-Scheming, Impossible Tasks (Coding, Flaky Tools). In particular, tasks in Repo-Sabotage and Impossible tasks are very close to real coding tasks that LLMs are being used for in the wild.
>
> We agree, however, that further validation against real deployment monitoring scenarios is an important next step. We see two key future directions here. First, we plan to expand the suite’s coverage so it better reflects the diversity of practical monitoring settings and failure modes. Second, because we intend to open-source at least part of the evaluation suite, we hope others can stress-test, extend, and validate these evaluations against additional practical settings. We view this paper as a first step toward a broader and continually improving set of monitorability evaluations.
>
> ___
> You also mentioned that the evaluations are somewhat specific to the class of models being evaluated (Model-Specific Eval Design). We agree that the evaluations are partly model-contingent, and that this is an important consideration when comparing across model families or capability levels. However, we would also like to note that your criticism could be applied to most control evaluations, where you need some level of propensity for a model to commit a bad action to evaluate if you can detect such actions. It’s further often true even for capability evaluations; for example, no one uses the GSM math evaluation anymore because it’s saturated and no longer useful in the context of modern LLMs.
>
> We acknowledge this is a real limitation and already address this in two ways. First, for cross-model comparisons, we drop evaluations that are invalid or insufficiently supported for any point in a figure, rather than averaging over different subsets of evaluations for different models, reasoning efforts, or training steps. This is intended to make within-figure comparisons fair by ensuring that all points are compared on the same subset of valid evaluations (this is described in detail in Appendix C.) Second, this is one reason we emphasize a broad suite spanning multiple evaluation types rather than relying on any single dataset or intervention. We intend to continually expand and update our open-sourced evaluation suite as model classes change, since a suite that is appropriate for one generation of models may become less informative for future systems if its interventions, process labels, or outcome properties no longer reflect how those systems behave.

---

> > ### Author Rebuttal · Reviewer_Wski · 2026-04-01
> >
> > Thanks for the response and providing useful context for my questions! I don't have further questions and continue to think that this is a good paper and should be accepted.

---

> > > ### Author Response · Authors · 2026-04-03
> > >
> > > Thank you so much!

---

### Official Review · Reviewer_kGnS · 2026-03-19

**Soundness:** 4
**Presentation:** 4
**Significance:** 4
**Originality:** 4
**Overall Recommendation:** 5
**Confidence:** 4

**Summary:**

This paper addresses the problem of evaluating how well chain-of-thought (CoT) monitoring can detect safety-relevant properties in reasoning models. The authors propose a formalized monitorability metric (g-mean²), three evaluation archetypes (intervention, process, and outcome-property), and a broad evaluation suite. They study scaling trends across pretraining size and inference-time compute, compare frontier reasoning models, and show that follow-up questions can improve monitorability.

**Compliance With Llm Reviewing Policy:**

Affirmed.

**Key Questions For Authors:**

The paper repeatedly highlights that monitorability improves at "medium to high reasoning effort" and that reasoning effort is a "deployment choice" and concrete lever for practitioners. However, the notion of reasoning effort appears to be operationalized differently across model families (API-level settings for OpenAI models,) Could the authors provide a more precise, model-agnostic definition of reasoning effort, or at minimum clarify how the different operationalizations are made commensurable when drawing cross-model conclusions? Without this, it is difficult to assess whether the scaling trends reported in Section 5 reflect a genuine property of reasoning effort or are confounded by model-specific implementation details. A clear answer here would strengthen my confidence in the generalizability of the paper's central practical recommendation.

**Limitations:**

Yes. The authors are commendably transparent about their limitations throughout the paper. They explicitly acknowledge the single-seed constraint in scaling experiments, the potential monitor bias from using an OpenAI model to evaluate predominantly OpenAI agents, label noise challenges in open-ended outcome-property evaluations, the stopgap nature of their filtering procedures for ShadeArena and Repo-sabotage, the limited generalizability of process evaluations beyond well-structured math/physics problems, and the fact that many training pressures on monitorability remain untested (e.g., deliberative alignment, indirect selection pressure from patching only detectable failures). The impact statement is minimal but reasonable given the nature of the work, which is fundamentally aimed at improving AI safety evaluation rather than introducing capabilities that could be misused.

**Strengths And Weaknesses:**

Soundness: The submission is technically sound — the g-mean² metric is rigorously motivated with clear demonstrations of F1's failure modes, the minimal-criterion TPR elegantly handles causal attribution under well-stated monotonicity assumptions, and the degeneracy evaluation suite provides strong evidence that monitors genuinely attend to CoT content rather than exploiting shallow heuristics. The main concerns are single-seed scaling experiments that make the monitorability tax estimates unreliable (the authors themselves note a mid-sized model is off-trend), and the stopgap label-noise filtering in ShadeArena/Repo-sabotage that may inflate recall by systematically excluding harder-to-detect ambiguous cases.

Presentation: The paper is exceptionally well-structured, with the progression from metric formalization to evaluation archetypes to empirical results to scaling trends to actionable improvements forming a coherent and compelling narrative, and Table 1's taxonomy alongside Figure 7's monitorable vs. unmonitorable CoT examples making abstract concepts immediately concrete and accessible. The work positions itself thoroughly against prior literature (faithfulness work, toy hint evaluations, misalignment benchmarks), clearly articulates how it generalizes each, and provides sufficient detail across the main text and appendices, including full monitor prompts, data construction procedures, and bootstrap methodology,

Significance: The paper addresses one of the most critical open questions in AI safety, whether CoT monitoring remains viable as models scale, and delivers actionable findings that practitioners can immediately use, including the monitorability tax tradeoff (smaller model at higher reasoning effort for better oversight) and follow-up questions as a concrete lever for improving monitoring in deployment. The evaluation suite and metric framework are likely to become standard infrastructure for safety cases relying on CoT monitoring, and the scaling trend analysis opens new research directions around preserving monitorability under training pressures, making the scope of impact both broad and appropriately urgent for the field.

Originality: The work's primary novelty lies in unifying scattered prior efforts (toy hint evaluations, individual misalignment benchmarks, faithfulness studies) into a principled evaluation framework with three complementary archetypes, and introducing the g-mean² metric adapted specifically for monitorability evaluation, which elegantly solves class-imbalance invariance problems that plague F1 while incorporating novel extensions like arm-specific TNR for intervention settings and minimal-criterion TPR for causal attribution. The process evaluation concept using constrained solution paths is a genuinely new evaluation primitive, and the empirical insights around scaling trends, the monitorability tax, and the finding that CoT access unlocks steeper compute-to-monitorability scaling provide novel understanding that clearly advances beyond what prior work has established.

---

> ### Author Rebuttal · Authors · 2026-03-31
>
> Thank you so much for your review! Throughout most of the paper, we standardized different operationalizations of reasoning effort by plotting monitorability against output token length (sum of CoT and final answer) or CoT token length, which are model-agnostic measures of reasoning effort. We will update Figure 6 to be a line plot where the x-axis is “Cot Tokens” instead of “increasing reasoning effort”).  We will update the x-axis of Figure 5 to be “Approx. Monitor Output Tokens” instead of “Monitor reasoning effort” (we unfortunately did not record the exact number of monitor output tokens for all evals and cannot recover it without rerunning all evals, which is very expensive). We hope this addresses your concern!
>
> Finally, we would like to share that will have published an open source version of our metrics code and the following datasets prior to the camera-ready paper deadline:
> * Intervention evals: GPQA, GPQA encoded, AIME, DAFT, Memory,  Scruples, and Sandbagging
> * Process evals: AIME-Math and GPQA-Physics
> * Outcome-property evals: Flaky-Tools (CharXiv subset), ShadeArena, and Impossible Coding Tasks (LiveBench subset)
>
> The other datasets rely on private or restricted data and will be omitted from the release.

---

> > ### Author Rebuttal · Reviewer_kGnS · 2026-04-04
> >
> > Thank you very much. I don't have further questions

---

> > > ### Author Response · Authors · 2026-04-05
> > >
> > > Thank you!

---

### Decision · Program_Chairs · 2026-04-30

**Decision:**

Accept (spotlight)

**Comment:**

The submission presents a very detailed study to quantify how well an LLM can be monitored by another LLM and how to conduct comparisons across models. Its insights into what model/training properties have a negative/positive effect on monitorability are an important contribution to the AI (safety) field. There is unanimous consensus that this work is well written, shows valuable results, and presents them in a great way. During the author-reviewer discussion, all questions and concerns of the reviewers were resolved. This paper will be a great addition to the ICML program.